# New Progress in the Molecular Regulations and Therapeutic Applications in Cardiac Oxidative Damage Caused by Pressure Overload

**DOI:** 10.3390/antiox11050877

**Published:** 2022-04-29

**Authors:** Xiaomeng Shi, Arin Dorsey, Hongyu Qiu

**Affiliations:** Center for Molecular and Translational Medicine, Institute of Biomedical Science, Georgia State University, Atlanta, GA 30303, USA; xshi8@student.gsu.edu (X.S.); adorsey17@student.gsu.edu (A.D.)

**Keywords:** antioxidants, heart, oxidative stress, pressure overload, reactive oxygen species

## Abstract

Chronic pressure overload is a key risk factor for mortality due to its subsequent development of heart failure, in which the underlying molecular mechanisms remain vastly undetermined. In this review, we updated the latest advancements for investigating the role and relevant mechanisms of oxidative stress involved in the pathogenesis of pressure-overload-induced cardiomyopathy and cardiac dysfunction, focusing on significant biological sources of reactive oxygen species (free radical) production, antioxidant defenses, and their association with the cardiac metabolic remodeling in the stressed heart. We also summarize the newly developed preclinical therapeutic approaches in animal models for pressure-overload-induced myocardial damage. This review aims to enhance the current understanding of the mechanisms of chronic hypertensive heart failure and potentially improve the development of better therapeutic strategies for the associated diseases.

## 1. Introduction

Oxidative stress, defined as the disturbed homeostasis between the production and detoxification of reactive oxidative species (ROS) in biological systems, has been considered a major pathological factor in many diseases. ROS or free radicals collectively refer to any chemical species containing unpaired electrons that highly increase the chemical reactivity of an atom or molecule, such as the hydroxyl radical, superoxide anion (O_2_^−^), nitric oxide (NO), NO-derived peroxynitrite, and transition metals, such as iron and copper. 

ROS can be generated in response to external electromagnetic radiation and can be produced by a myriad of biochemical processes within the body that involves the incomplete reduction of molecular oxygen. Many studies have established that ROS production at low levels and specific sites acts as a crucial physiological mediator in both intracellular and extracellular signal transduction, cell proliferation, vascular health, host defense, and the epigenetic and retrograde control of gene expression. 

On the contrary, ROS can be highly damaging to organisms at high concentrations and participate in the pathogenesis of human illnesses, such as cardiovascular, aging, and neurodegenerative diseases. In the meantime, all cells possess delicately orchestrated networks of antioxidant defenses to maintain ROS homeostasis in normal physiological conditions. If this balance is disturbed, tissue damage may occur and thus contribute to the development of various pathological processes. The primary and most abundant source for ROS generation inside the cell is mitochondrial oxidative metabolism. The electron transport chain deploys radical species to transfer electrons via oxidative phosphorylation to generate ATP [1,2]. 

The myocardium has the most significant number of mitochondria per cell and relies almost entirely on aerobic respiration. Therefore, there is no doubt that oxidative stress coupled with redox signaling plays a vital role in cardiac pathophysiology. Additionally, there are several dedicated sources in the myocardium that could also produce ROS, such as membrane-bound NADPH oxidases (Nox), uncoupled nitric oxide synthases, xanthine oxidase (XO), monoamine oxidases, and cytochrome P450 oxidase, which are most commonly characterized in controlling cardiac redox signaling [3,4]. 

Reciprocally, the heart has an integrated complex antioxidant defense system comprised of enzymatic and non-enzymatic antioxidants to counteract ROS accumulation and neutralize ROS into nontoxic molecules. Enzymatic antioxidants form the first line of defense against ROS, such as superoxide dismutase (SOD), catalase (CAT), glutathione (GSH), glutathione peroxidases (GPx), paraoxonase (PON), and thioredoxin (Trx). In contrast, non-enzymatic antioxidants constitute the second line of defense by interrupting free radical chain reactions [5]. 

Pressure overload in the heart, for instance, caused by hypertension or aortic stenosis, induces structural and functional remodeling in response to the increased workload. As initially compensated concentric hypertrophy gradually progresses to decompensated eccentric hypertrophy under chronic pressure overload, the heart undergoes a series of pathological remodeling processes in terms of contractile dysfunction, electrophysiological abnormalities, and metabolic inflexibility [6,7]. 

Pressure overload is considered to stimulate cardiomyocyte growth, known as hypertrophy, which leads to enlargement and thickening of ventricular walls accompanied by increased synthesis and deposition of collagen and extracellular matrix proteins, termed fibrosis [8]. Pressure overload also triggers a transient perivascular inflammatory response in the myocardium and a wave of apoptosis, necrosis, capillary rarefaction, and phagocytosis of cardiomyocytes [9,10,11]. Prolonged pressure overload induces increased oxidative stress, mitochondrial dysfunction, and coexisting myocyte autophagy [12]. 

In contrast, chronic pressure-overload-induced oxidative stress leads to excessive mitochondrial DNA damage, defective mitochondrial ATP generation, overproduction of free radicals, and impaired contractile dysfunction due to ROS-mediated nonenzymatic modifications of proteins central to excitation-contraction coupling [13,14]. Moreover, ROS has been suggested to activate numerous kinases and transcriptional factors involved in hypertrophic signaling pathways in the myocardium while serving as pro-fibrotic mediators in myocardial fibrosis and a vital apoptotic stimulant for cardiomyocytes [15,16]. 

These primary pathological responses, free radicals, and their sources implicate secondary injury responses under chronic pressure overload, which progressively aggravates myocardial function and eventually leads to heart failure. Myocardial antioxidants are believed to act concertedly to reduce ROS-induced cell damage through diverse avenues, such as inhibiting metal-catalyzed ROS formation, promoting the antioxidant generation, upregulating antiapoptotic protein expression, and scavenging ROS precursors and free radicals [17]. 

Under pressure overload or ischemic insults, the endogenous antioxidant mechanisms in the heart tend to be overwhelmed by ROS overflow, resulting in diminished antioxidant capacity [4]. Therefore, targeting antioxidants and antioxidant signaling pathways represent a new direction of therapeutic strategies for treating ROS-associated heart diseases. A copious amount of evidence has recently emerged demonstrating the cardioprotective effects of dietary interventions, such as multivitamins and pharmacological applications with specific antioxidants or free radical scavengers in counteractive defense and prevention of myocardial ROS stress in animal models of pressure overload and heart failure [18,19,20].

This review summarizes the recent experimental evidence on the molecular mechanisms regulating oxidative stress in pressure-overloaded hearts. We also discuss emerging therapeutic approaches that can effectively improve pressure-overload-induced adverse cardiac remodeling by reducing excessive oxidative stress, thus, providing a greater perspective on novel treatment strategies for pressure-overload-induced heart conditions. 

## 2. Oxidative Stress in Response to Cardiac Pressure Overload 

Oxidative stress has been extensively characterized for its detrimental effects on the functional and structural integrity of the heart in various diseases. This review mainly highlighted the new progress during the past five years in the regulatory mechanisms of ROS production and antioxidative defense under the condition of pressure overload. In addition, studies have shown that pressure overload induces a significant metabolic remodeling in the myocardium, which eventually results in a deficient energy supply in the heart and subsequent cardiac dysfunction [21,22]. Emerging evidence indicates a tight association between cardiac oxidative stress and cardiac metabolic remodeling in response to the pathological insult of pressure overload; thus, we specifically updated recent discoveries to improve our understanding of this appreciated topic. 

### 2.1. ROS Production in Experimental Models of Pressure Overload

#### 2.1.1. Newly Identified Proteins Associated with Mitochondrial Ros Production and Related Regulating Mechanisms

Since mitochondria are the major primary source of ROS production in the heart under cardiac stress, numerous studies have been conducted to explore the molecular mechanisms regulating ROS production in the heart’s mitochondria. Here, we included recent discoveries of mitochondrial proteins or mitochondrial redox signaling-associated proteins exerting protective or detrimental effects on the heart under pressure overload. 

Recent studies have identified several mitochondrial proteins regulating pressure-overload-induced oxidative stress. For example, mitochondrial GTPases 1 (MTG1) protein expression has been found to be significantly increased in failing human hearts and pressure-overloaded mouse hearts after aortic banding (AB) [23]. MTG1 overexpression reversed the pressure-overload-induced loss of mitochondrial complex I, II, and IV activities, preserving the integrity of the mitochondrial respiratory complex against pressure overload. 

In contrast, MTG1 deficiency exacerbated pressure-overload-induced mitochondrial dysfunction and cardiomyopathy in MTG1 knockout (KO) mice, which was significantly attenuated with the treatment of N-acetyl-L-cysteine, a powerful antioxidant [23]. These results indicate that MTG1 could potentially serve as a novel therapeutic target for protecting mitochondrial function and reducing oxidative stress in pressure-overloaded hearts. Reciprocally, cardiac-specific conditional knockout of the mitochondrial 18 kDa translocator protein (TSPO) has been reported to attenuate pressure-overload-induced impairment in complex I activity. 

Furthermore, TSPO decreased mitochondrial membrane potential (ΔΨm) and lowered excessive superoxide production in the heart of TSPO KO mice subjected to transverse aortic constriction (TAC) [24]. These findings suggested a potential pharmacological target of inhibiting TSPO expression to substantially preserve cardiac mitochondrial function and oxidative stress balance, preventing or treating pressure overloaded-induced cardiac injury. 

In addition, some proteins were found to be involved in the cardiac protection against pressure-overload-induced cardiac hypertrophy and dysfunction through regulating mitochondrial redox signaling. For instance, cardiac-specific overexpression of valosin-containing protein (VCP), an ATPase associated with various activities, has been found to ameliorate persistent pressure-overload-induced maladaptive hypertrophic remodeling and contractile dysfunction in transgenic (TG) mice subjected to TAC [25]. 

The further genomic analysis identified a marked downregulation of cardiac mitochondrial genes involved in oxidative stress, especially several supernumerary subunits of complex I, III, and IV in VCP TG mice under pressure overload than the sham controls, accompanied by remarkably lowered oxidative markers in stressed hearts. These results suggest that VCP could protect the heart against pressure-overload-induced oxidative injury [25]. 

#### 2.1.2. Nox-Associated ROS Production in a Pressure-Overloaded Heart

Unlike the mitochondria that generate ROS as a secondary effect, Nox enzymes produce ROS as a primary function [26]. The phagocyte NADPH oxidase itself (Nox2/gp91phox) and its homologs are now referred to as the Nox family of NADPH oxidases, including Nox1, Nox3, Nox4, Nox5, and dual oxidase 1/2 (Duox1/2) where the Nox family has been implicated in a wide range of pathological processes in the heart [27]. Among the seven isoforms, Nox5 and Duox1/2 require calcium-dependent activation. Therefore, they have been previously characterized as superoxide-generating pro-contractile Nox isoforms that crosslink calcium and redox signaling in mammalian vascular contraction [28]. Nox2 is the most extensively studied subtype, and it is coexpressed with Nox4 as the two main isoforms in cardiomyocytes.

Multiple studies have demonstrated that Nox’s play a negative role in the heart subjected to pressure overload insult. The expression and functional significance of Nox5 in cardiac hypertrophy were recently studied using human cardiac tissues, transgenic mice, and neonatal rat cardiomyocytes [29]. In failing human hearts, cardiac expression of Nox5 was significantly increased compared with normal donor hearts, and Nox5 (V2) was the only isoform whose mRNA expression was increased. 

Nox5-derived ROS stress was associated with altered MAPK signaling in the pressure-overloaded heart. Furthermore, Nox5 overexpressing transgenic (Nox5-Tg) mice subjected to either TAC or Angiotensin II (AngII) infusion exhibited worsened cardiac hypertrophy and myocardial contractile dysfunction compared with TAC or AngII-treated wild-type (WT) mice. TAC also caused a significant increase in NADPH-induced O_2_^−^ production in the hearts of Nox5-Tg mice than their WT littermates which was consistent with observations made in vitro. 

Aside from increased Nox5 expression in failing human hearts, Nox2 expression was also upregulated in human cardiac tissues from patients with dilated cardiomyopathy [30]. Genetic deletion of Nox2 was observed to suppress pressure-overload-induced myocardial interstitial fibrosis, hypertrophy, and oxidative stress in TAC-treated Nox2 KO male mice compared with WT-TAC controls. Such inhibitory effects were accompanied by a consistent downregulation of TAC-induced activation of MAPKs, including ERK1/2, JNK, and p38, in the hearts of Nox2 KO mice. These results suggest that inhibiting Nox2 and Nox5 expression in the myocardium would oppositely counteract cardiac oxidative stress in response to pressure overload. 

In addition, multiple studies demonstrate genes or proteins regulating pathological cardiac remodeling under pressure-overload through Nox-mediated ROS production. For example, cardiac expression of ubiquitin-specific protease 2 (USP2) was significantly diminished in pressure-overloaded mouse hearts after TAC. 

Overexpression of USP2 via tail injection of recombinant adeno-associated virus serotype 9 (AAV9) in TAC-treated male mice has been reported to attenuate pressure-overload-induced ventricular dysfunction, cardiac fibrosis, inflammation, and overall oxidative stress compared with AAV9 control-treated counterparts [31]. Specifically, USP2 overexpression could markedly reduce elevated Nox2, Nox4, and p22phox at the mRNA level in pressure-overloaded hearts, indicating an unprecedented regulatory role of USP2 in mediating oxidative stress and Nox protein expression during cardiac pressure overload. 

β3-adrenergic receptor (AR) stimulation was found to alleviate pressure-overload-induced cardiac hypertrophy, fibrosis, and cardiomyocyte apoptosis while improving cardiac dysfunction [32]. In contrast, treatment with β3-AR inhibitor (SR) increased ROS production and upregulated Nox activity in pressure-overloaded hearts of TAC-treated male mice. Yet, the administration of β3-AR agonist (BRL) abolished pressure-overload-induced ROS production upregulation and Nox subunit expression of p22phox, Nox2, p47phox, p67phox, and Rac1 [32]. β3-AR stimulation-mediated cardioprotection was connected to the activation of neuronal nitric oxides synthase (nNOS).

Furthermore, genetic deletion of low-density lipoprotein receptor (LDLr) by using adeno-associated virus vector serotype 8 (AAV8) has been found to counteract pressure-overload-induced cardiac hypertrophy, cardiomyocyte apoptosis, diastolic dysfunction, and interstitial and perivascular myocardial fibrosis in TAC-treated female LDLr^−/−^ mice [33]. 

LDLr deletion dramatically attenuated pressure-overload-induced elevation in plasma lipid peroxidation malondialdehyde (MDA) level, plasma XO activity, nitro-oxidative stress markers, and Nox2 and Nox4 protein expression levels in the myocardium [33]. LDLr deletion remarkably increased the myocardial SOD protein level and its plasma activity to promote antioxidant responses, thus, protecting the pressure-overloaded heart from oxidative injury in female LDLr^−/−^ mice with mild hypercholesterolemia. 

### 2.2. Antioxidant Response in Experimental Models of Pressure Overload

The heart has a complex and multilevel network of antioxidant defense to neutralize ROS and detoxify ROS accumulation to counteract ROS overproduction from various resources and biological activities under extreme stress conditions. Major endogenous enzymatic defenses, such as SOD, CAT, GSH, GPx, PON, and Trx, form the front line of protection against ROS-mediated injury in all living organisms. Here, we updated recently identified new targets that were involved in cardiac antioxidant responses in experimental models of pressure overload and their relevant regulatory mechanisms.

#### 2.2.1. Nuclear Factor-Erythroid Factor 2-Related Factor 2 (Nrf2)-Associated Regulators

Nrf2, also known as nuclear factor erythroid-derived 2-like 2, is a basic leucine zipper protein that functions as a critical transcription factor regulating the expression of antioxidant proteins that protect against oxidative damage. Emerging evidence has revealed a vital role of Nrf2 activation in protecting against maladaptive cardiac remodeling by regulating the transcriptional activities of a myriad of antioxidant response element (ARE)-dependent genes. These ARE-dependent genes encode detoxification enzymes and cytoprotective proteins. 

As such, Nrf2/ARE pathway activation mediates the induction of NAD(P)H quinone oxidoreductase 1 (NQO1) and heme oxygenase-1 (HO-1) to elicit powerful detoxication and elimination abilities to remove electrophiles and reactive oxidants, thus, conferring cellular resistance to oxidative stress [34,35]. Kelch-like ECH-associated protein 1 (Keap1) serves as the primary negative regulator of NRF2, and the Keap1-Nrf2 pathway is considered the principal cytoprotective response to oxidative and electrophilic stresses [35]. 

Previously, Nrf2 deletion has been found to aggravate pressure-overload-induced myocardial hypertrophy, source substantial mitochondrial damage, promote cardiomyocyte apoptosis, and contractile dysfunction in Nrf2^−/−^ mice compared to pressure overload WT hearts after aortic constriction [36]. More interestingly, an Nrf2-regulated gene program was increased during chronic pressure overload but suppressed in the heart of Nox4^−/−^ mice. 

Apart from this, cardiac-specific Nox4 overexpression-induced protective qualities against chronic pressure overload were significantly attenuated in Nox4TG/Nrf2^−/−^ mice with worsened contractile dysfunction and hypertrophy compared with Nox4TG/Nrf2^+/+^ mice that exhibited considerably less adverse cardiac remodeling. These findings revealed a Nox4-dependent Nrf2 upregulation in the heart against chronic pressure overload, and induction of this cytoprotective pathway may represent a new therapeutic approach.

Recent studies have identified manifold molecules in the heart that modulate cardiac antioxidative responses through the Nrf2-mediated signaling against pressure-overload-induced oxidative stress. One example is peroxiredoxin-1 (Prdx1), an efficient antioxidant enzyme previously upregulated in failing mouse hearts, which was studied in pressure-overloaded hearts of TAC-treated male mice [37]. 

Myocardial Prdx1 overexpression via AAV9-Prdx1 injection significantly inhibited pressure-overload-induced pathological cardiac remodeling post-TAC, including myocardial hypertrophy, dysfunction, fibrosis, inflammation, and oxidative stress compared with control groups. Mechanistically, Prdx1 overexpression significantly diminished TAC-induced cardiac ROS production, and activity of both MDA and Nox, with improved SOD and CAT activity. 

This also reversed pressure overload-induced decrease in Nrf2 and HO-1 levels and increased nuclear translocation of Nrf2 from the cytosol [37]. Furthermore, knockdown of Nrf2 by Nrf2 siRNA was found to abolish the antihypertrophic effect of Prdx1 overexpression and abrogate its antioxidant defense, suggesting that cardiac-specific Prdx1 overexpression may provide cardioprotection by modulating the Nrf2/HO-1 antioxidant pathway.

Another example involves Sestrin2 (Sesn2), a highly conserved stress-inducible metabolic protein and known free radical repressor, whose expression was significantly elevated in the hypertrophic hearts of AB-treated male mice compared to sham-operated control [38]. The observed cardiomyocyte-specific Sesn2 overexpression attenuated pressure-overload-induced cardiac dysfunction, hypertrophy, and fibrosis in Sesn2 TG mice compared with WT control and markedly improved the decreased expression of Nrf2, HO-1, SOD1, and SOD2. It also significantly inhibited increased p67 and Nox2 expression and accumulated lipid peroxidation in Sesn2 TG mice compared to the WT controls. This study indicates that Sesn2 may function as a positive regulator of Nrf2 activation in the heart, serving as a potential target for antioxidant therapy against pressure-overload-induced oxidative stress.

Correlatively, tripartite motif containing-21 (TRIM21), an interferon-inducible E3 ligase, may function as a negative regulator of Nrf2 signaling in pressure-overloaded hearts. Genetic ablation of TRIM21 fully preserved cardiac contractility under severe TAC in TRIM21 knockout mice compared with TAC-treated control littermates [39]. Deletion of TRIM21 eminently attenuated TAC-induced histone H2A variants H2AX and elevated NQO1 expression. In addition, TRIM21-mediated p62 ubiquitylation at residue K7 negatively regulated p62-promoted antioxidant responses via the p62/Nrf2 axis. These results indicated that loss of TRIM21 could provide cardioprotection in pressure-overloaded hearts through enhanced antioxidant responses. 

#### 2.2.2. Newly Discovered Regulators of the Antioxidant Defense System in the Heart Primarily through Modulating First-Line Defense Antioxidants

Anoctamin-1 (ANO1), a voltage-gated calcium-activated anion channel protein highly expressed in epithelial cells, appears to be significantly reduced in myocardial tissues following TAC-induced pressure overload [40]. Inversely, ANO1 overexpression significantly attenuates pressure-overload-induced maladaptive cardiac hypertrophy, fibrosis, and cardiomyocyte apoptosis in ANO1 TG mice hearts compared to WT mice counterparts [40]. More than that, ANO1 overexpression markedly improved decreased SOD1 expression and reduced the increased 4-hydroxynonenal (4-HNE) content in cardiac tissues of ANO1 TG mice, revealing a novel cardioprotective effect of ANO1 utilizing antioxidant responses. 

In contrast to a strongly positive role of ANO1 against pressure overload in the heart, mixed lineage kinase 3 (MLK3), a member of the MAP3K family, has been previously characterized as a negative regulator in protecting against cardiomyocyte injury [41,42,43]. Recently, MLK3 depletion by injecting an AAV-MLK3 vector (AAV^MLK3−^) into TAC-treated mice (TAC + AAV^MLK3−^) was found to significantly alleviate pressure-overload-induced cardiac dysfunction, fibrosis, pyroptosis, and ferroptosis more than control mice (TAC + AAV^NC^) [43,44]. The expression of antioxidant response proteins, including cystine/glutamate transporter (xCT or SLC7A11), glutathione peroxidase 4, and ferritin heavy chain 1, were markedly increased in MLK3 KO TAC mice when compared to WT mice. 

Beyond this, TAC + AAV^MLK3−^ exhibited significantly improved SOD and GSH levels but substantially decreased MDA levels in the heart compared with TAC + AAV^NC^. Cyclooxygenase 2, JNK, and p53, whose expressions mediate oxidative stress-induced ferroptosis, were significantly reduced in MLK3 KO TAC mice, revealing additional cardioprotective effects involving MLK3 inhibition against pressure overload.

Additionally, the deletion of P66Shc, a mitochondrial redox signaling protein that produces hydrogen peroxide (H_2_O_2_), has been reported to ameliorate TAC-induced increase in myocardial NADPH level, superoxide production, and lipid peroxidation in p66Shc KO mice compared with WT littermates. Contrarily, p66Shc overexpressed mice exhibited an aggravated phenotype [45]. GSH and SOD were markedly increased in p66Shc KO mice but suppressed by p66Sh overexpression compared with WT littermates, respectively. 

In addition, TAC increased protein levels of phosphodiesterase type 5 (PDE5), and a series of ventricular oxidative stress markers, such as 4-HNE, were attenuated in p66Shc KO mice and augmented in p66Shc overexpressed mice. Treatment with SOD mimetic M40401 or PDE5 inhibitor sildenafil could significantly decrease ventricular oxidative stress markers. This study demonstrated a regulatory role of p66Shc in pressure-overload-induced oxidative stress through downstream of SOD and PDE5, thus, bringing potential therapeutic value for pressure-overload-induced heart failure. 

In summary, these studies from animal models provided new evidence further supporting the contribution of oxidative stress in the pathogenesis of cardiomyopathy and dysfunction caused by pressure overload and the development of heart failure. These new discoveries also highlighted the significance of potential therapeutic approaches to maintain the homeostasis between ROS production and detoxification by targeting both mechanisms. The newly identified potential targets are summarized in Figure 1.

### 2.3. Cardiac Metabolic Remodeling-Associated Oxidative Stress and Antioxidative Regulation upon Pressure Overload 

Perturbations in myocardial energy metabolism have been considered causative for cardiac pathogenesis, culminating in detectable tissue injury and malfunction. Unlike other tissues in the body, the heart prioritizes long-chain fatty acids as the primary cardiac energy source to generate ATP through mitochondrial beta-oxidation. Alternatively, glucose is less favored by the myocardium. Under the stress of pressure overload, the energy-deficient heart switches from fatty acid oxidation (FAO) to the more oxygen-efficient glucose oxidation [7]. 

Such metabolic shift is adaptive in the early stage of hypoxic cardiac tissue since higher oxygen efficiency could ameliorate ischemia but not in the chronic pressure-overloaded heart. Many studies indicated that metabolic remodeling induced by chronic pressure overload is associated with mitochondrial damage and impaired oxidative capacity that inevitably results in excessive ROS formation [21,46,47,48]. This section discusses new findings regarding the regulatory effects of cardiac metabolic alterations on cardiac oxidative stress and their potential therapeutic values in pressure-overloaded hearts. These updated studies are summarized in Figure 2. 

Fatty acid metabolism: Recently, free fatty acid receptor 4 (Ffar4) deficiency was discovered to aggravate pressure-overload-induced cardiac dysfunction following exacerbated development of concentric hypertrophy in Ffar4KO male mice subjected to chronic TAC [49]. Treatment with Ffar4 agonist TUG-891 specifically elevated intracellular production of oxylipin 18-hydroxyeicosapentaenoic acid (18-HEPE), a fatty acid metabolite, in WT cardiac myocytes but not Ffar4 KO cardiac myocytes. Moreover, TUG-891 pretreatment reduced H_2_O_2_-induced oxidative stress in WT cardiac myocytes whose expression of HO-1 was uniquely upregulated. 

18-HEPE was barely detected in Ffar4KO high-density lipoprotein, suggesting an indispensable role of Ffar4 in its basal production and a novel cardioprotective role via inducing the 18-HEPE expression and mitigating oxidative stress [49]. In addition, deficiency of perilipin 5 (PLIN5), a lipid droplet-associated protein that maintains the balance between lipolysis and lipogenesis, was found to irritate TAC-induced myocardial hypertrophy and heart failure in Plin5-null mice compared to WT littermates [50]. This protein expressed in highly oxidative tissues could inhibit TAC-induced excessive FAO. 

In contrast, its knockdown significantly elevated levels of MDA and ROS in the hearts of Plin5-null mice subjected to TAC. The activity of the antioxidant enzyme SOD, on the contrary, was significantly decreased, suggesting a higher oxidative burden due to excessive FAO in the TAC-treated Plin5-deficient heart. This study provided insights into how cardiac lipid metabolic dysfunction contributed to the pathogenesis of cardiac pressure overload, in which enhanced FAO and subsequent oxidative stress played a central role. 

Glucose metabolism: Sodium-glucose cotransporter 1 (SGLT1), a major glucose carrier in the heart, has been lately linked to increased myocardial nitro-oxidative stress in rats subjected to TAC-induced chronic pressure overload compared with sham control [51]. The expression of SGLT1, with nitrosative stress marker nitrotyrosine (NT) and oxidative stress marker 4-HNE, was upregulated in TAC-treated rats. Moreover, cardiac SGLT1 expression was strongly and positively correlated with Nox4 protein expression, implying a potentially harmful role of SGLT1 in the heart failure models [51]. 

Another freshly identified protein is TP53-induced glycolysis and apoptosis regulator (TIGAR), which possesses an intrinsic fructose bis-phosphatase activity that lowers glycolysis. One study showed that, compared to sham control, TAC induced a progressive upregulation of TIGAR expression in WT mice. On the other hand, TIGAR deficiency in TIGAR-KO TAC myocardium eliminated significant ROS damage by promoting its secondary antioxidant property by upregulating the pentose phosphate pathway to produce NAPDH, which suggests a novel therapeutic approach to counteract pressure-overload-induced oxidative injury [52]. 

Additionally, cardiac-specific deletion of sirtuin 1 (Sirt1), a critical metabolic sensor of glucose and lipid metabolism, has been found to induce mild left ventricular (LV) systolic dysfunction and increased propensity to produce ROS in cardiac mitochondria of Sirt1 KO mice (Sirt1^ciKO^) with augmented protein carbonylation, an irreversible oxidative protein modification [53]. 

Sirt1^ciKO^ mice exhibited tremendous hypertrophic growth, LV dysfunction, and significantly decreased myocardial mitochondrial respiration rates in response to TAC-induced pressure overload compared with the sham group. Moreover, TAC-Sirt1^ciKO^ mice exhibited dramatically reduced citrate synthase and cytochrome c oxidase activities, suggesting worsened mitochondrial dysfunction in Sirt1-deficient hearts under pressure overload [53]. 

Ketone body metabolism: WT TAC-treated hearts under chronic pressure overload intrinsically upregulated 3-hydroxybutyrate dehydrogenase 1 (BDH1) expression and expectedly increased ketone body utilization, both of which were observed in Bdh1 overexpressed TG hearts with and without being subjected to TAC [54]. 8-Hydroxy-2′-deoxyguanosine (8-OHdG), a critical endogenous biomarker of oxidative DNA damage induced by TAC, was substantially reduced in Bdh1 TG hearts. 

In vitro, adenovirus-mediated Bdh1 overexpression was found to markedly reduce intracellular ROS production and cardiomyocyte apoptosis in β-hydroxybutyrate-treated H9c2 cells while enhancing the mRNA expression of crucial antioxidants and antioxidative enzymes, such as metallothionein 2 (MT2), CAT, SOD, and forkhead box O3A. As baseline mRNA expression of MT2 and CAT did not differ significantly between Bdh1 TG mice and WT mice, these results indicate that Bdh1 overexpression-mediated cardioprotection through the enhanced antioxidative defenses may emerge in the condition of sufficient ketone bodies in response to significant energy deprivation, such as after TAC.

Amino acid metabolism: An increased plasma concentration of homocysteine has long been considered to independently predict major adverse cardiac events, such as the development of chronic heart failure, especially in females [55]. Therefore, selective homocysteine-lowering gene transfer using an E1E3E4-deleted hepatocyte-specific adenoviral vector that expresses cystathionine-β-synthase (AdCBS) has been studied for its potential benefits in pressure-overloaded hearts of female Ldlr^−/−^Cbs^+/−^ mice treated with TAC [56]. 

TAC Ldlr^−/−^Cbs^+/−^ mice fed with a methionine-rich, folate-deficient diet to establish mild hyperhomocysteinemia presented with worsened cardiac hypertrophy, diastolic dysfunction, interstitial fibrosis, and apoptosis than control TAC mice fed with a standard chow and diet AdCBS TAC mice. Furthermore, significantly reduced plasma concentrations of GSH and cysteine while GPx and SOD activities detected in diet TAC mice were markedly elevated in control TAC mice and diet AdCBS TAC mice. TAC-induced elevated plasma lipid peroxidation and reactive nitrogen species formation were significantly lower in control TAC mice and diet AdCBS TAC mice than in diet TAC mice. Collectively, CBS ablation could counteract TAC-induced oxidative stress and augment antioxidant defenses, which confer beneficial effects on pressure-overload-induced cardiomyopathy.

In recent years, there has also been rapid progress in branched-chain amino acids (BCAA) metabolism involved in cardiac metabolic remodeling. Germ-line knockout of Krüppel-like factor 15 controlled BCAA catabolic gene protein phosphatase 2C in mitochondria (PP2Cm) has been associated with promoting cardiac dysfunction and accelerating cardiac hypertrophy in TAC-treated PP2Cm KO mice when compared with WT TAC-treated control [57]. Branched-chain α-ketoacid accumulation due to PP2Cm knockdown created a defective BCAA catabolism that could directly inhibit complex I mediated respiration, significantly increasing myocardial superoxide production and total protein oxidation in TAC-treated PP2Cm-KO hearts. This study highlighted a new role of BCAA catabolism in mediating oxidative stress in pressure-overloaded hearts.

## 3. Newly Developed Antioxidative Therapeutic Approaches against Cardiac Pressure Overload

In recent years, there has been vast research investigating potential pharmacological targets of oxidative stress that can effectively provide cardioprotection. Many studies have identified additive cardioprotective effects against pressure-overload-induced oxidative stress from previously approved medications prescribed for treating other diseases, e.g., the repurposed pharmacological agents. In addition, many natural organic compounds extracted from herbs and species have demonstrated powerful antioxidant effects to counteract pressure-overload-induced oxidative stress. Aside from pharmacological approaches, nonpharmacological interventions, such as calorie restriction, play an important therapeutic role in pressure-overload-induced cardiac dysfunction. This updated information is presented in Table 1.

### 3.1. Repurposed Pharmacological Agents

Several approved drugs have been identified to improve pressure-overload-induced cardiac injury in animal models outside of their initial medical indication. These repurposed pharmacological agents offer significant advantages to developing an entirely new drug for treating pressure overload-associated cardiac diseases.

Raloxifene, a previously approved drug for the prevention and treatment of osteoporosis in postmenopausal women, has been found to considerably reduce pressure-overload-induced myocardial hypertrophy, fibrosis, inflammation, and attenuate cardiac dysfunction in TAC mouse hearts through intragastric administration compared with control-treated counterparts [58]. Raloxifene could significantly improve SOD2 expression and SOD activity while lessening inducible nitric oxide synthase expression and mitophagy-related protein, including Pink1, Parkin, and Bnip3, to the baseline level in interleukin 6 (IL-6)-treated H9c2 myoblasts. These results revealed inhibitory effects of raloxifene on either TAC or IL-6 induced oxidative stress and autophagy dysfunction that possibly involved the IL-6/STAT3 signaling, thus, likely providing cardioprotection towards the pressure-overloaded heart.

Fasudil is a potent Rho-kinase inhibitor and vasodilator that has been used to treat cerebral vasospasm since it was discovered. Rats subjected to TAC exhibited significantly decreased cardiac antioxidant activities of SOD, CAT, and GPx with increased MDA content. Furthermore, pretreatment with fasudil via subcutaneous injection could dramatically boost antioxidant activities and reduce MDA levels in the hearts of TAC rats compared with vehicle-treated counterparts [59]. Fasudil administration could heavily reverse elevated expression of Nrf2 and HO-1 while drastically decreasing Keap1 expression. Fasudil-induced increase in cardiac iron content was identified as potentially triggering mild oxidative stress that, in turn, activated the Nrf2 signaling, which subsequently upregulated the activities of antioxidant enzymes to suppress oxidative stress [59].

Celecoxib, a nonsteroidal anti-inflammatory drug, exhibited cardioprotection from pressure-overload-induced hypertrophic remodeling, contractile dysfunction, cardiomyocyte apoptosis, and interstitial fibrosis in rats subjected to abdominal aortic constriction (AAC) [60]. Celecoxib treatment could inhibit AAC-induced upregulation in cardiac MDA levels while rescuing decreased gene expressions of Nrf2-mediated antioxidant enzymes, HO-1, NQO-1, and Nrf itself. Increased expressions of the negative regulators Nrf2 and Keap1 in the hypertrophic heart were found to be significantly reduced by celecoxib treatment. These findings showed strong evidence regarding the antioxidant property of celecoxib in the hypertrophic heart, which could serve as a potential therapeutic option for pressure-overload-induced cardiac hypertrophy [60].

Sacubitril/Valsartan (LCZ696) is the first agent approved as an angiotensin receptor neprilysin inhibitor (ARNI) to treat patients who have chronic heart failure with reduced ejection fraction [63]. Administration of LCZ696 via oral gavage has been reported to significantly improve cardiac function, attenuate cardiac hypertrophy, and reduce cardiac fibrosis in TAC-treated mice [61]. There were less myocardial accumulation of superoxide and peroxide derivatives, such as NT and 4-HNE, in LCZ696-treated TAC mice compared with in vehicle-treated TAC mice.

Moreover, TAC caused downregulation in the expression levels of antioxidants manganese superoxide dismutase (MnSOD) and sirtuin3 (Sirt3) were markedly reversed by LCZ696 treatment. In contrast, Sirt3 knockdown aggravated phenylephrine (PE)-induced oxidative stress in cultured cardiomyocytes and abolished LCZ696 conferred cardioprotection. This study demonstrated that LCZ696 could mediate an antihypertrophic effect in TAC-treated mice by upregulating the MnSOD/Sirt3 pathway against myocardial oxidative stress [61]. Oral administration of LCZ696 has also been found to attenuate persistent pressure-overload-induced LV fibrosis to a greater extent in AAC-treated male rats than those treated with valsartan alone [62].

Moreover, LCZ696 was found to reduce mitochondrial superoxide levels greater than valsartan or sacubitril alone in ventricular myocytes isolated from pressure-overloaded rat hearts and AngII-treated primary adult ventricular myocytes, revealing a powerful antioxidative effect against cardiac oxidative stress. In addition, treatment with MitoTEMPO, a potent mitochondria-targeted antioxidant, nearly abolished AngII-stimulated increase in total cellular superoxide production in these myocytes, confirming the mitochondrial origin of ROS production in response to pressure overload [62].

### 3.2. Naturally Derived Organic Extracts

A great deal of research has explored potential treatments for pressure-overload-related heart diseases using natural, organic components derived from plants. Here, we summarized the latest findings in this field.

Oridonin, an active diterpenoid extracted from the Chinese herbal medicine Rabdosia was currently found to inhibit pressure-overload-induced myocardial hypertrophy and fibrosis while attenuating cardiac oxidative stress in mice subjected to AB [64]. Oridonin treatment via oral gavage could significantly decrease the expression of cardiac gp91phox, p67phox, and SOD2 at both mRNA and serum levels in AB-treated mice. Myocardial expression of HO-1 was restored in oridonin-treated AB mice, while 4-HNE levels were primarily diminished. In addition, overall myocardial ROS production and MDA levels were decreased. Still, SOD and GPx levels were increased in AB-treated mice under oridonin treatment, revealing a powerful antioxidant property of this compound against pressure-overload-induced oxidative stress.

Apocynin, a natural organic compound structurally related to vanillin, has been isolated from various plant sources and serves as the most employed Nox inhibitor. Previously, apocynin has been able to ameliorate pressure-overload-induced myocardial hypertrophy in rats subjected to AAC [66]. Recent progress revealed that apocynin treatment could effectively attenuate pressure-overload-induced increases in Nox activity in the heart tissues of AAC-treated rats compared to vehicle-treated control. In addition, apocynin treatment could reverse declined myocardial SOD activity and effectively reduce the increased O_2_^−^ and MDA levels [65]. This study revealed apocynin-mediated cardioprotection through suppressing oxidative stress and boosting antioxidant responses.

Carnosic acid (CA), a natural phenolic terpenoid isolated from Rosmarinus officinalis previously characterized as an effective lipid peroxidation inhibitor, was recently investigated for its potential effects on cardiac remodeling in mice subjected to aortic banding-induced pressure overload [67]. Orally administered carnosic acid was observed to decrease myocardial MDA production and NADPH movement while increasing SOD activity in pressure-overloaded mice against vehicle-treated AB groups. CA treatment further attenuated pressure-overload-induced increases in Nox2 and Nox4 expression and noticeably suppressed 4-HNE expression. Mechanistically, the protective effects of carnosic acid against pressure-overload-induced oxidative stress were associated with the downregulation of the AKT/GSK3β/Nox4 signaling pathway.

Stachydrine treatment, a constituent of the Chinese herb Leonurus heterophyllus sweet, via oral gavage alleviated pressure-overload-induced cardiac hypertrophy, contractile dysfunction, and excessive autophagic activity in TAC-treated rats [68]. In vitro, the AngII-induced dramatic increase in ROS production was significantly blocked by stachydrine pretreatment in H9c2 cells. Stachydrine treatment abolished the elevated gp91phox and p67phox expression along with the increased phosphorylation of p47phox and gp91phox colocalization in AngII-treated H9c2 cells. The above findings demonstrated that stachydrine could effectively halt Nox2 cytosolic subunit translocation to the membrane and subsequent overactivation in H9c2 cells in response to AngII-induced oxidative stress.

Nobiletin (NOB), a polymethoxy flavonoid isolated from citrus peels, has been discovered to attenuate pressure-overload-induced cardiac hypertrophy, fibrosis, endoplasmic reticulum stress, and cardiomyocyte apoptosis while improving cardiac dysfunction in AB-treated mice via oral gavage [69]. NOB also significantly restored the decreased SOD1 concentration level and suppressed increased Nox2 and Nox4 expression in addition to 4-HNE levels in heart tissues of NOB-treated AB mice.

Through intraperitoneal injection, treatment with astragaloside IV (AS-IV), an active substance isolated from Astragalus membranaceus, has been shown to improve pressure-overload-induced cardiac dysfunction, reverse cardiomyocyte hypertrophy, and myocardial mitochondrial damage in TAC-treated male mice [70]. AS-IV administration significantly reduced the increased H_2_O_2_ content in myocardial mitochondria and abolished the augmented cardiomyocyte apoptosis in AS-IV-treated TAC mice. Such protective effects of AS-IV were associated with the regulation of sentrin-specific protease 1 expression.

Once more delivered by intraperitoneal injection, treatment with cardamonin (Cam), a chalconoid extracted from several plants, such as Alpinia katsumadai, has been demonstrated to significantly alleviate pressure-overload-induced cardiac hypertrophy, fibrosis, cardiomyocyte apoptosis, and myocardial dysfunction in TAC-treated mice [71]. TAC-induced cardiac 4-HNE expression was significantly reduced following Cam administration in TAC mice. In vitro, H9C2 cells treated with Ang II, H_2_O_2_, and Nox4 considerably decreased in SOD and GSH content along with an increased MDA level indicative of direct oxidative damage. These pathological changes were remarkably reversed after Cam treatment. This study is the first to provide evidence on the potential cardioprotective effect of Cam in heart failure due to chronic pressure overload [71].

Aucubin (AUB), an iridoid glycoside in plants, established itself to suppress PE-induced cardiomyocyte hypertrophy in vitro and significantly attenuate chronic pressure-overload-induced cardiac hypertrophy, fibrosis, inflammation, and cardiac dysfunction in AB-treated male mice via intraperitoneal injection [72]. Moreover, AUB treatment was observed to inhibit PE-induced ROS generation and mRNA expression of P67phox. In contrast, AUB increased mRNA expression of SOD and GPx in PE-treated H9c2 cells, suggesting a protective role against oxidative stress in vitro. Consistently, AUB-treated AB mice exhibited decreased lipid peroxidation as indicated by reduced 4-HNE levels compared with vehicle-treated control. Further investigation revealed that AUB-induced cardioprotection against pressure overload relied on increased nNOS expression [72].

Hispidulin, a naturally occurring flavonoid in different plants, was shown to attenuate pressure-overload-induced myocardial hypertrophy and improve cardiac dysfunction in male mice subjected to descending AB through intraperitoneal injection in comparison to their vehicle-treated AB mice counterparts [73]. In addition, hispidulin restored pressure overload suppressed mitochondrial oxidative phosphorylation, and preserved mitochondrial structure in the hypertrophic mouse hearts while improving impaired mitochondrial respiration and ATP production. Moreover, hispidulin significantly increased the expressions of SOD1, MnSOD, and CAT in the hearts of AB-treated mice at the mRNA level. Hispidulin-conferred cardioprotection against myocardial hypertrophy, mitochondrial inflexibility, and oxidative stress in response to pressure overload was achieved by activating its downstream target, Sirt1 [73].

### 3.3. Natural Organic Compounds

In addition to natural organic ingredients derived from plants, many other natural organic compounds or food supplements have been investigated for their potential value in protecting the heart from pressure overload. For example, barbarum polysaccharides (LBPs) isolated from a traditional Chinese herb *Lycium barbarum* L., now a popular food supplement, have been recently evaluated for their antioxidative effects in the hearts of Wistar rats subjected to abdominal aorta banding (AAB) [74]. Commercial ready-to-use LBPs extracts administered orally to AAB-treated mice could significantly lower plasma MDA levels than control AAB mice, suggesting a robust antioxidant effect of LBPs against pressure-overload-induced oxidative stress in the failing heart [74].

Lycopene, a type of organic pigment especially abundant in tomatoes and a powerful natural antioxidant, has been displayed to suppress pressure-overload-induced myocardial hypertrophic responses and cardiac dysfunction in AB-treated mice [75]. Lycopene-treated AB mice exhibited increased SOD1 expression and less-abundant superoxide production at the mRNA level in myocardial tissues compared with control AB mice. In vitro, lycopene-pretreated cardiomyocytes significantly lowered mitochondrial ROS production and remarkably enhanced ARE activity in response to PE stimulation compared with vehicle-treated counterparts.

Furthermore, lycopene pretreatment could improve the PE-inhibited expression of HO-1, SOD1, and CAT in cardiomyocytes at the mRNA level, indicating that both ARE activity and ARE-mediated antioxidant enzyme expression were upregulated by lycopene. Lycopene-mediated antioxidant actions were associated with downregulated phosphorylation of ERK1/2, p38, and JNK, as well as Akt (Thr308) and GSK3β of the pro-hypertrophic signaling pathway [75].

Intraperitoneal injection of fisetin, a plant flavonol and dietary antioxidant, has been reported to inhibit pressure-overload-induced myocardial hypertrophy while mitigating cardiac dysfunction in male mice subjected to descending AB [76]. In vitro, fisetin treatment could suppress hypertrophic cardiomyocyte growth and the re-emergence of fetal gene expression in PE-treated cultured cardiomyocytes without exerting cytotoxicity. Moreover, fisetin treatment reduced ROS production in the hearts of sham mice by about 20% and AB-treated mice by about 30%. In addition, fisetin was found to reduce both PE-induced and baseline ROS production in vitro, suggesting that fisetin could protect the heart from pressure overload through attenuating cardiac oxidative stress [76].

Vitamin D (VD) administered via oral gavage in TAC-treated male mice has been reported to significantly inhibit pressure-overload-induced cardiac hypertrophy, fibrosis, and inflammation while improving consequent cardiac dysfunction compared with the vehicle-treated control [77]. Likewise, VD-3-treated TAC mice exhibited a dose-dependent inhibition of cardiomyocyte apoptosis, myocardial superoxide production, and mRNA expression of Nox2, Nox4, and p22phox compared to vehicle-treated TAC mice [77].

Intravenous injection of irisin, an isolated hormone from mouse skeletal muscle, in TAC-treated male rats significantly lowered TAC-induced increase in mRNA expression of Nox2 and XO while increasing SOD1 and plasma GPx levels compared with vehicle-treated TAC rats [78]. In addition, the upregulated expressions of p-Akt, p-mTOR, and p-GSK3β observed in rats after TAC was significantly suppressed by irisin treatment, suggesting an inhibitive role of irisin on the Akt signaling system to confer cardioprotection against pressure overload [78].

Inhibition of cathelicidin-related antimicrobial peptide (CRAMP) by transfecting AB-treated mice with recombinant AAV9-shCRAMP via intraperitoneal injection has been reported to ameliorate pressure-overload-induced cardiac hypertrophy, inflammation, and oxidative stress [79]. Specifically, CRAMP treatment notably enhanced the antioxidant activity of SOD2 and GPx while reducing ROS production. Additionally, Nox2 and Nox4 expression levels in hypertrophic hearts of CRAMP-deficient AB mice were diminished. The antioxidative stress effect of CRAMP treatment was mediated through the activation of toll-like receptor 9 (TLR9) of the TLR9/AMPKα signaling pathway in pressure-overloaded failing hearts [79].

Oral administration of Qindan capsule (QC), a compound used in traditional Chinese medicine to treat hypertension, has been found to significantly attenuate pressure-overload-induced cardiac hypertrophy and fibrosis while improving the impaired cardiac function and myocardial ultrastructure in TAC-treated male mice compared to vehicle-treated control [80]. Moreover, QC treatment significantly minimized TAC-induced increase in cardiac expression levels of 8-OHdG, MDA, and 15-isoprostane F2t with decreased AngII concentrations in the heart tissues.

Intragastric administration of taurine, an organic compound widely prevalent in animal tissues, has exhibited abilities to prevent TAC-induced deterioration of cardiac function in TAC-treated male mice and significantly attenuate cardiomyocyte hypertrophy and fibrosis compared with vehicle-treated control [81]. Furthermore, taurine treatment significantly lowered the pressure-overload-induced increase in cardiomyocyte apoptosis, ROS production, and MDA expression while enlarging SOD expression in the hearts of the TAC group. Taurine-induced beneficial effects were then determined to be mediated by activating the Sirt1-p53 pathway [81].

### 3.4. Potential Chemical Compounds with Antioxidant Property

Many chemical compounds that potentially interact with ROS signaling have been identified as promising therapeutic targets to treat pressure-overload-induced heart diseases in the future.

Exogenous alpha-calcitonin gene-related peptide (a-CGRP) administration was shown to enhance cardiac function while attenuating myocardial hypertrophy and fibrosis in TAC-treated mice [82]. Treatment of a-CGRP significantly decreased pressure-overload-induced myocardial overexpression of 4-HNE, 8-OHdG, and MDA. In contrast, a-CGRP restored the reduced total GSH. The observed antioxidative effects and cardioprotection against pressure overload were likely mediated by Sirt1 and AMPK [82].

Treatment by the hippo pathway inhibitor, XMU-MP-1, which targets the expression of mammalian Ste-20-like kinase 1 and mammalian Ste-20-like kinase 2, was found to attenuate myocardial hypertrophic responses and reduce cardiac apoptosis and fibrosis. XMU-MP-1 improved cardiac function in TAC-treated mice [83]. XMU-MP-1 addition significantly enhanced cell survival in H_2_O_2_^−^ treated rat neonatal cardiac myocytes, suggesting a cardioprotective effect in cardiomyocytes against oxidative stress.

A small Wnt inhibitor molecule called Wnt-C59 constrained porcupine required for Wnt palmitoylation and was discovered to significantly attenuate pressure-overload-induced hypertrophic responses while improving cardiac dysfunction and enhancing the survival of adult male mice subjected to TAC [84]. Wnt-C59 treatment incredibly reduced upregulated ROS production, partially blocked GPx activity, markedly reversed the increased MDA levels, and decreased SOD activity in pressure-overloaded hearts.

Treatment with mitoquinone (MitoQ), a robust mitochondria-targeted antioxidant, via oral gavage in AAC-treated male mice has improved pressure-overload-induced LV contractile dysfunction and mitochondrial depolarization. MitoQ preserved disrupted mitochondrial networks [85]. MitoQ treatment could also effectively reduce MDA levels in the AAC myocardium, likely by modulating the redox-sensitive Plscr4-miR-214 axis.

Isolevuglandins (isoLGs) adducts have recently been discovered to modify cardiac proteins into neoantigens that subsequently engage in CD4^+^ T-cell receptor activation in pressure-overloaded hearts [86]. Orally administered isoLGs scavenger 2-hydroxybenzylamine (2-HOBA) and its less reactive isomer 4-HOBA could significantly inhibit TAC-induced excessive LV ROS production, suggesting potentially beneficial effects of 2-HOBA as an effective antioxidant and scavenger of free radicals in pressure-overloaded hearts [87].

### 3.5. Calorie Restriction

On top of these medications, studies have shown that non-pharmacological interventions play a beneficial role in response to pressure overload. For example, calorie restriction (CR) has been studied in mouse hearts subjected to pressure overload following ascending aortic constriction (AAC) [88]. Mice fed with 60% of the given calorie intake two weeks before and after AAC significantly reduced LV hypertrophy, interstitial fibrosis, and impaired LV relaxation compared with control AAC mice fed in ad libitum (AL). Furthermore, CR treatment attenuated the AAC-induced increase in both 8-OHdG and mitochondrial content of lipid hydroperoxide in myocardial tissues.

Excessive Nox-dependent and mitochondrial superoxide production detected in the AL + AAC group were significantly reduced in the CR + AAC group. On the contrary, cardiac GPx and SOD activities were substantially increased in the CR + AAC group but not in the AL + AAC group. It is important to note that the initial two weeks of CR before AAC did not bring any significant differences in cardiac morphological, functional, or redox changes between AL and CR intake groups, suggesting that short-term CR could be applied as a valuable strategy to manage pressure-overload-induced cardiac injury [88].

Recently, preconditioning with two weeks of dietary restriction (DRPC) has also been reported to alleviate pressure-overload-induced maladaptive cardiac remodeling in the left ventricle of DRPC-treated male mice subjected to 2 weeks of AB with the AL + AAC group [89]. AAC-induced increases in 8-OHdG, mitochondrial lipid hydroperoxide, and excessive Nox-dependent and mitochondrial superoxide production detected in the hypertrophic LV myocardium of the AL + AAC group were significantly suppressed in the AL DRPC + AAC group. This study demonstrated the attenuating effects of short-term DRPC to counteract pressure-overload-induced cardiac oxidative stress [89].

## 4. Conclusions and Future Perspectives

Cardiac pressure overload is a common pathological condition in clinical settings, and this condition results in the concentric hypertrophy of cardiac muscles with contractile dysfunction that eventually progresses to heart failure, thus, contributing to high morbidity and mortality rates among elderly patients. Recent experimental studies in animal models, using either genetic modification or pharmacologic intervention on target genes, demonstrated that pressure-overload-induced oxidative stress plays a vital role in adverse cardiac remodeling and the subsequent progression to heart failure.

Cardiac oxidative stress is tightly associated with a maladaptive metabolic remodeling in response to pressure overload, further complicating the pathogenesis and consequences of pressure overload in associated heart diseases. Thus, numerous studies have explored potential therapeutic strategies to inhibit oxidative stress by enhancing antioxidant responses while suppressing ROS production.

Emerging therapeutic strategies involve repurposed pharmacological agents, naturally occurring organic compounds and ingredients, synthetic chemical compounds, and calorie restriction. These new advances offer different perspectives for understanding the pathogenesis of pressure overload-associated heart diseases and provide novel evidence for developing strategies to combat chronic heart failure. The therapeutic targets summarized above potentially unlocked more treatment options that could develop in the future.

For example, Nox isoforms have been extensively reported to be upregulated in the stressed heart, fueling ROS production. Thus, there has begun a quest for developing selective Nox inhibitors, such as Nox-specific peptidic inhibitors, Nox2ds-tat, and small-molecule inhibitors [90]. In addition, activating the Nrf2 pathway to induce its cardiac protective properties through herbal Nrf2 activators has gained some preclinical progress [91].

Despite recent progress in elucidating the mechanisms of pressure-overload-induced oxidative stress in the heart, many aspects remain undetermined and require further investigation. First, although various methods for assessing ROS production are available for in vitro studies, direct detection of ROS in vivo has presented challenges due to its very short half-life and high reactivity. Further endeavors are required to develop methods for in situ, real-time detection of ROS in vivo.

Second, although animal studies using transgenic or knockout mouse models have provided strong evidence for the regulatory mechanisms of pressure-overload-induced cardiac injury, much work remains to be done to translate these experimental observations into clinical use. Third, current animal models of experimental pressure overload are generated by surgically restricting aortic blood flow to increase heart afterload, which only somewhat mimics the complex structural and functional changes that develop in response to pressure overload in human hearts.

Therefore, there is a constant need to develop more reliable and informative animal models that can better mimic pressure-overload-induced heart failure’s multifactorial and combinatorial pathogenicity. Fourth, the therapeutic effects of newly developed drugs or compounds in animal experiments are rather unspecific or indirectly related to antioxidative efforts against pressure-overload-induced cardiac oxidative stress.

Although targeting antioxidant mechanisms has been considered a promising cardioprotective strategy to combat adverse effects of excessive ROS, a plethora of clinical trials have produced mixed results regarding their efficacy [92]. For example, there has been conflicting evidence regarding the actions of Nox4 in the myocardium as both Nox4 inhibition and cardiac-specific overexpression have shown beneficial effects in the stressed heart via different mechanisms. In addition, research targeting ROS-coupled metabolic disturbances to thwart ROS-induced damage through modifications of related metabolism regulatory pathways has also yielded mixed results.

Nrf2, a previously characterized master regulator of antioxidative responses, has recently been incriminated in inducing myocardial damage and dysfunction, revealing a dark side in the heart [93]. These inconsistencies suggest that manipulating antioxidant agents or related pathways are not necessarily effective and can even be harmful, bringing formidable challenges to successfully implementing pharmacological concepts into clinical practice. Thus, further and more thorough investigations are required to provide more direct evidence and precise molecular mechanisms of their therapeutic actions that could potentially reach a better clinical outcome.

Although this review focuses on oxidative stress and antioxidant defenses involved in pressure-overload-induced cardiac injury, oxidative stress with shared mechanisms of action is substantially implicated in a range of cardiovascular conditions. For instance, multiple Nox isoforms previously identified to be variably elevated in hypertensive animal models were comprehensively reviewed. Nox5 has been most recently highlighted as a crucial mediator of vascular oxidative stress, and Nox4 plays an especially important role in salt-sensitive hypertension [94].

Additionally, mitochondrial-targeted antioxidants, such as the aforementioned MitoQ, have been shown to reduce hypertension and hypertension-associated organ damage [95,96]. The overlapped ROS mechanisms identified in different pathological situations could motivate the development of universally applicable therapies against a spectrum of ROS-associated cardiovascular conditions.

## Figures and Tables

**Figure 1 antioxidants-11-00877-f001:**
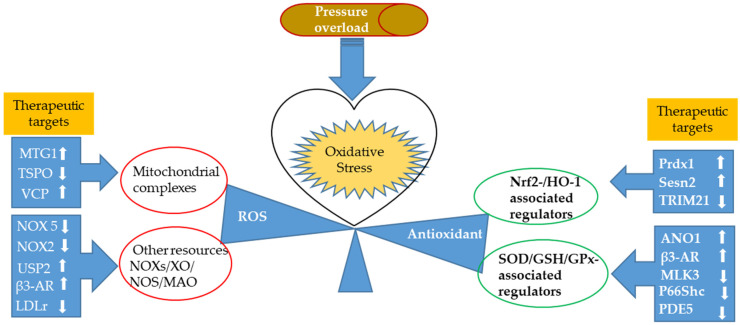
The updated potential therapeutic targets involved in attenuating cardiac oxidative stress resulted from pressure overload. There have been newly identified diverse therapeutic targets that regulate cardiac oxidative stress induced by pressure overload in animal models. Cardiac-specific overexpression of mitochondrial GTPases 1 (MTG1), valosin-containing protein (VCP), and conditional knockout of mitochondrial 18 kDa translocator protein (TSPO) could preserve cardiac mitochondrial quality and integrity, reduce ROS production, and attenuate myocardial injury under pressure overload. Genetic deletion of NADPH oxidase 5 (Nox5), NADPH oxidase 2 (Nox2), and low-density lipoprotein receptor (LDLr) suppress Nox activity and subsequent Nox-dependent ROS production to protect pressure-overloaded hearts from oxidative injury. Stimulation of ubiquitin-specific protease 2 (USP2) and β3-adrenergic receptor (AR) also exert substantial Nox inhibition to counteract cardiac oxidative stress. From the perspective of enhancing antioxidative defense, overexpression of peroxiredoxin-1 (Prdx1), Sestrin2 (Sesn2), and genetic ablation of tripartite motif containing-21 (TRIM21) could upregulate nuclear factor erythroid-2 related factor 2/heme oxygenase-1 (Nrf2/HO-1) pathway that subsequent active antioxidants defend against cardiac oxidative stress induced by pressure overload. Anoctamin-1 (ANO1) overexpression, deletion of mixed lineage kinase 3 (MLK3), and Src homology/collagen (Shc) adaptor protein (P66Shc) could upregulate major antioxidants in the heart, such as superoxide dismutase (SOD), glutathione (GSH), and glutathione peroxidase (GPx), to protect the stressed heart from oxidative injury.

**Figure 2 antioxidants-11-00877-f002:**
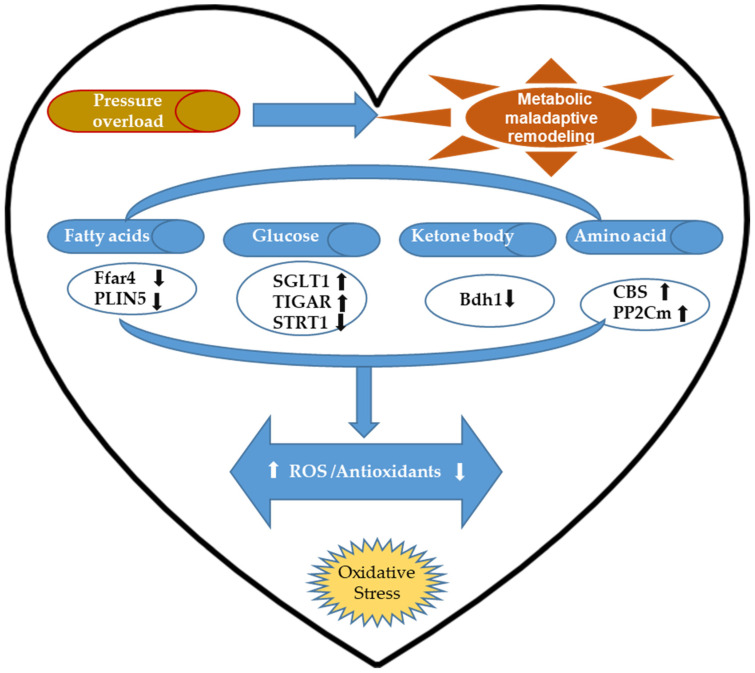
The updated potential regulation of metabolic remodeling on cardiac oxidative stress implicated in pressure-overload-induced cardiac injury. Cardiac oxidative stress induced by pressure overload is tightly associated with decreased mitochondrial oxidative capacity and maladaptive metabolic remodeling that shifts from predominantly using fatty acid oxidation to relying more on glucose, amino acids, and ketone metabolism under hypoxic conditions. Free fatty acid receptor 4 (Ffar4) deficiency attenuates excessive ROS production and multifaceted cardiac damage caused by stress overload, whereas perilipin 5 (PLIN5) deficiency does the opposite. Sodium-glucose cotransporter 1 (SGLT1) expression in pressure-overloaded hearts is positively correlated with increased oxidative and nitrosative stress. Moreover, cardiac-specific deletion of sirtuin 1 (Sirt1) promotes myocardial sensitivity to pressure-overload-induced cardiac damage. In contrast, TP53-induced glycolysis and apoptosis regulator (TIGAR) deficiency alleviates cardiac oxidative injury following pressure overload insult. 3-hydroxybutyrate dehydrogenase 1 (BDH1) overexpression reduces ROS production and enhances antioxidant expression in pressure-overloaded hearts while improving ketone utilization. Protein phosphatase 2C in mitochondria (PP2Cm) deficiency causes defective branched-chain amino acids (BCAA) catabolism and inhibits mitochondrial respiration accumulation of myocardial superoxides in the pressure-overloaded hearts. Cystathionine-β-synthase (CBS) deficiency lowers pressure-overload-induced plasma lipid peroxidation and reactive nitrogen species formation while improving plasma superoxide dismutase (SOD), glutathione (GSH), and glutathione peroxidase (GPx).

**Table 1 antioxidants-11-00877-t001:** The newly developed therapeutic approaches against cardiac pressure overload.

	Therapeutic Approaches	Route of Administration	Animal Model	Targeted Antioxidative Mechanisms in the Heart	References
Repurposed pharmacological agents	Raloxifene	Oral gavage	TAC	↑SOD expression and activity and ↓iNOS expression via IL-6/STAT3 signaling	[58]
	Fasudil	Subcutaneous injection	TAC	↑Nrf2/HO-1 pathway and↑SOD, CAT, and GPx activities	[59]
	Celecoxib	Oral gavage	AAC	↑Nrf2-mediated HO-1, NQO-1, and ↓MDA	[60]
	Sacubitril/Valsartan (LCZ696)	Oral gavage	TAC	↓Superoxide and peroxide derivatives and ↑MnSOD and Sirt3	[61,62,63]
Naturally derived organic extracts	Oridonin	Oral gavage	AB	↓gp91phox, p67phox and ROS production; ↑HO-1, SOD, and GPx	[64]
	Apocynin	Voluntary oral ingestion	AAC	↓Nox activity, O_2_^−^, and MDA; ↑SOD activity	[65,66]
	Carnosic acid (CA)	Oral gavage	AB	↓AKT/GSK3β/Nox4 signaling; ↑SOD activity	[67]
	Stachydrine	Oral gavage	TAC	↓gp91phox and p67phox expression, p47phoxphosphorylation, and p47phox/gp91phox colocalization	[68]
	Nobiletin (NOB)	Oral gavage	AB	↑SOD1 concentration; ↓Nox2 and Nox4 expression and 4-HNE levels	[69]
	Astragaloside IV (AS-IV)	Intraperitoneal injection	TAC	↓H_2_O_2_ content	[70]
	Cardamonin (Cam)	Intraperitoneal injection	TAC	↓4-HNE and MDA; ↑SOD and GSH content	[71]
	Aucubin (AUB)	Intraperitoneal injection	AB	↓ROS generation, P67phox expression, and lipid peroxidation; ↑SOD, GPx, and nNOS expression	[72]
	Hispidulin	Intraperitoneal injection	AB	↑SOD1, MnSOD, and CAT expression	[73]
Natural organic compounds	L. barbarum L. polysaccharides (LBPs)	Oral gavage	AAB	↓plasma MDA levels	[74]
	Lycopene	Oral gavage	AB	↑ARE activity and ARE-mediated HO-1, SOD1, and CAT expression; ↓ROS production	[75]
	Fisetin	Intraperitoneal injection	AB	↓ROS production	[76]
	Vitamin D (VD)	Oral gavage	TAC	↓superoxide production, Nox2, Nox4, and p22phox expression	[77]
	Irisin	Intravenous injection	TAC	↓Nox2 and XO; ↑SOD1 and plasma GPx	[78]
	Cathelicidin-related antimicrobial peptide (CRAMP)	Intraperitoneal injection	AB	↑SOD2 and GPx activity; ↓Nox2 and Nox4 expression	[79]
	Qindan capsule (QC)	Oral gavage	TAC	↓8-OHdG, MDA, and 15-isoprostane F2t	[80]
	Taurine	Oral gavage	TAC	↓ROS production and MDA expression; ↑SOD expression	[81]
Potential chemical compounds with antioxidant property	Alpha-calcitonin gene-related peptide (a-CGRP)	Subcutaneous injection	TAC	↓4-HNE, 8-OHdG, and MDA; ↑total GSH	[82]
	XMU-MP-1	Intraperitoneal injection	TAC	↑enhanced cell survival against H_2_O_2_	[83]
	Wnt-C59	Oral gavage	TAC	↓ROS production and lipid peroxidation; ↑GPx and SOD activity	[84]
	Mitoquinone (MitoQ)	Oral gavage	AAC	↓MDA levels via redox-sensitive Plscr4-miR-214 axis	[85]
	2-hydroxybenzylamine (2-HOBA)	Oral gavage	TAC	↓ROS production	[86,87]
Non-pharmacological interventions	Calorie restriction (CR)	Voluntary oral ingestion	AAC	↓8-OHdG, mitochondrial content of lipid hydroperoxide, Nox-dependent and mitochondrial superoxide production; ↑GPx and SOD activities	[88]
	Dietary restriction preconditioning (DRPC)	Voluntary oral ingestion	AAC	↓8-OHdG, mitochondrial content of lipid hydroperoxide, Nox-dependent, and mitochondrial superoxide production	[89]

TAC: Transverse aortic constriction; SOD: Superoxide dismutase; iNOS: Inducible nitric oxide synthase; IL-6: Interleukin 6; Nrf2: Nuclear factor erythroid-derived 2-like 2; HO-1: Heme oxygenase-1; CAT: Catalase; GPx: Glutathione peroxidases; AAC: Ascending aortic constriction; NQO1: Quinone oxidoreductase 1; MDA: Malondialdehyde; MnSOD: Manganese superoxide dismutase; Sirt3: Sirtuin3; AB: Aortic banding; O_2_^−^: Superoxide anion; Nox4: NADPH oxidase 4; SOD1: Superoxide dismutase type 1; Nox2: NADPH oxidase 2; 4-HNE: 4-hydroxynonenal; GSH: Glutathione; H_2_O_2_: Hydrogen peroxide; ROS: Reactive oxygen species; nNOS: Neuronal nitric oxides synthase; AAB: Abdominal aorta banding; ARE: Antioxidant response element; XO: Xanthine oxidase; SOD2: Superoxide dismutase 2; 8-OHdG: 8-hydroxy-2′-deoxyguanosine; ↓: downregulation; ↑: upregulation.

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
