# Peer review of "New Progress in the Molecular Regulations and Therapeutic Applications in Cardiac Oxidative Damage Caused by Pressure Overload"

_antioxidants, 2022, doi:10.3390/antiox11050877_

Round 1
Reviewer 1 Report
The manuscript comprehensively summarized the new discoveries on reactive oxygen species (ROS) production and antioxidant mechanism of defense focusing on pressure-overload-induced myocardial damage. Generally, the review is well organized and there are just few things to highlight before acceptance. The followings are some comments to this review.
In the Introduction section, a description of chronic pressure overload is missing and a better connection between the disease and ROS/antioxidants is needed. In addition, for a better understanding for general readers, a description of antioxidants and their role in normal and pathological conditions would be helpful.
The two drawings presented by the authors in the manuscript are their own drawing / diagram. Authors should include reproductions of drawings already published by other authors summarizing the most relevant studies they describe in the submitted review.
A table summarizing the preclinical therapeutic approaches in animal models of pressure-overload-induced myocardial damage could make this article more useful.
Reviewer 2 Report
Shi and colleagues review the role of oxidative stress in the pathogenesis of pressure overload-induced cardiomyopathy and cardiac dysfunction. The authors summarise recent findings which have identified new ROS producing sources and antioxidant components, and discuss the latter’s association with maladaptive metabolic remodelling. Potential interventions for attenuating pressure overload-induced cardiac dysfunction are also discussed. The authors may consider to include brief paragraphs based on the following comments.
- Could there be overlap (or differences) in the ROS/antioxidant pathways between pressure overload-induced heart failure models, and hypertension models (e.g., Dahl salt–sensitive rats), as both models are used to study chronic hypertensive heart disease, yet their molecular signatures are different. If the pathways are indeed different, this may influence the intervention that is being developed; as such, a particular therapy may not be universally applicable to all patients.
- While oxidative stress is indeed a key driver of pathological cardiac remodelling and dysfunction, the clinical testing of antioxidants has yielded mixed results. The authors are kindly requested to refer to the following review article (PMID: 33675957), which discusses these inconsistencies in detail.
Reviewer 3 Report
Review article titled (New Progress in Oxidative Stress in Cardiac Pressure Overload and the Relevant Therapeutic Targets: Reactive Oxygen Species 3
and Antioxidants) by Shi et al. This is a well organized and comprehensive review that adds knowledge to the field. I have some comments for improvement:
1- The title: should be revised especially the second part. What is the new progress" mentioned in the title?
In addition: the second part does not add more value
2-Also mentioned in abstract"This progress would enhance the current state of knowledge r": what was this progress?
3- The style of writing needs revision as many unnecessary words appear in the sentences starting from the title, abstract...etc. the removal of these unnecessary words is necessary to improve the manuscript and make the paper easy to be read.
4- For the therapeutic targets: please give the examples for the therapeutic options based on them.
Author Response
A point-by-point response to Reviewer 3:
We sincerely thank the reviewer for the positive comments and also appreciate the constructive suggestions. We have addressed each of the comments and have revised the manuscript accordingly. We believe that the revised manuscript has significantly improved.
A point-by-point response is provided below, and the corresponding changes are presented in the revised manuscript and highlighted in blue.
1- The title: should be revised especially the second part. What is the new progress" mentioned in the title? In addition: the second part does not add more value
Response: We have revised the title to reflect the manuscript's contents more precisely as “New Progress in the Molecular Regulations and Therapeutic Applications in Cardiac Oxidative Damage Caused by Pressure Overload”.
2-Also mentioned in abstract"This progress would enhance the current state of knowledge r": what was this progress?
Response: We have revised the abstract to correct this confusing piece of writing.
3- The style of writing needs revision as many unnecessary words appear in the sentences starting from the title, abstract...etc. the removal of these unnecessary words is necessary to improve the manuscript and make the paper easy to be read.
Response: We have carefully checked the writing throughout the manuscript and removed the unnecessary words as suggested, and revised the sentences to be more concise.
4- For the therapeutic targets: please give the examples for the therapeutic options based on them.
Response: We have added examples for the therapeutic options in lines 772-779, as follows:
The therapeutic targets summarized above potentially unlocked more treatment options that could develop in the future. For example, Nox isoforms have been extensively reported to be upregulated in the stressed heart, fueling ROS production. Thus, there has started a quest for developing selective Nox inhibitors such as Nox-specific peptidic inhibitors, Nox2ds-tat, and small-molecule inhibitors[90]. In addition, activating the Nrf2 pathway to induce its cardiac protective property through herbal Nrf2 activators has gained some preclinical progress[91].
Round 2
Reviewer 2 Report
The authors have addressed all my comments.
Reviewer 3 Report
thanks for the revision